# Modulation of Cellular MicroRNA by HIV-1 in Burkitt Lymphoma Cells—A Pathway to Promoting Oncogenesis

**DOI:** 10.3390/genes12091302

**Published:** 2021-08-24

**Authors:** Beatrice Relebogile Ramorola, Taahira Goolam-Hoosen, Leonardo Alves de Souza Rios, Shaheen Mowla

**Affiliations:** Division of Haematology, Department of Pathology, Faculty of Health Sciences, University of Cape Town, Observatory 7925, Cape Town 7700, South Africa; beatrice.ramorola@gmail.com (B.R.R.); taahira.goolamhoosen@uct.ac.za (T.G.-H.); leonardo.rios@uct.ac.za (L.A.d.S.R.)

**Keywords:** MicroRNA, miR-200c, Burkitt lymphoma, HIV-1, non-Hodgkin lymphoma, ZEB

## Abstract

Viruses and viral components have been shown to manipulate the expression of host microRNAs (miRNAs) to their advantage, and in some cases to play essential roles in cancer pathogenesis. Burkitt lymphoma (BL), a highly aggressive B-cell derived cancer, is significantly over-represented among people infected with HIV. This study adds to accumulating evidence demonstrating that the virus plays a direct role in promoting oncogenesis. A custom miRNA PCR was used to identify 32 miRNAs that were differently expressed in Burkitt lymphoma cells exposed to HIV-1, with a majority of these being associated with oncogenic processes. Of those, hsa-miR-200c-3p, a miRNA that plays a crucial role in cancer cell migration, was found to be significantly downregulated in both the array and in single-tube validation assays. Using an in vitro transwell system we found that this downregulation correlated with significantly enhanced migration of BL cells exposed to HIV-1. Furthermore, the expression of the ZEB1 and ZEB2 transcription factors, which are promotors of tumour invasion and metastasis, and which are direct targets of hsa-miR-200c-3p, were found to be enhanced in these cells. This study therefore identifies novel miRNAs as role players in the development of HIV-associated BL, with one of these miRNAs, hsa-miR-200c-3p, being a candidate for further clinical studies as a potential biomarker for prognosis in patients with Burkitt lymphoma, who are HIV positive.

## 1. Introduction

MicroRNAs (miRNAs) are a class of small noncoding RNA molecules that play an essential role in gene regulation post-transcriptionally [1,2]. In their mature form, miRNAs are approximately 18–22 nucleotides long and are incorporated into the RNA-induced silencing complex (RISC) where they can act as guides, leading the complex to the 3′- UTR region of messenger RNA (mRNA). Upon complementary binding to the target mRNA, it is either targeted for degradation or translation is inhibited [3,4]. miRNAs are critical for normal cellular function and what has emerged clearly in the last decade is that aberrant miRNA expression is associated with many diseases, including malignancies. This is likely a result of amplification and/or deletion of specific genomic regions, with the deregulated miRNA having either an oncogenic or tumour suppressor role, affecting one or several of the cancer hallmark events [5,6].

Non-Hodgkin lymphomas (NHLs) represent a heterogeneous group of malignancies, which originate from lymphoid haematopoietic tissue, with the majority being of B-cell origin [7]. Quite early on in the acquired immune deficiency syndrome (AIDS) epidemic, an association between NHLs and AIDS was established, leading to the current classification of several NHL subtypes as being associated with HIV infection [8]. This includes Burkitt lymphoma (BL), a highly aggressive B-cell NHL characterized by the genetic hallmark *c-MYC* rearrangement with the immunoglobulin gene loci, and t(8;14) (q24;q32) being the most frequently observed translocation [9,10,11]. BL is one of the most common NHLs affecting HIV-positive individuals, and in developing countries, which carry the highest HIV burden worldwide, patients typically present with late-stage disease requiring aggressive treatment [12,13]. Importantly, recent evidence has emerged in support of HIV and HIV-1 encoded proteins having direct oncogenic effects, particularly in the development of BL, and therefore the pathobiology of HIV-associated cancers is unique and distinct from cancers of the same type that do not develop within an HIV-positive background [14,15,16,17,18]. This is likely to include alterations in miRNA expression. Oncogenic viruses such as the Epstein–Barr virus (EBV) and the Kaposi’s sarcoma-associated herpesvirus (KSHV), both of which are closely associated with HIV infection and the development of lymphoproliferative diseases, have been shown to alter the expression of a wide array of cellular miRNAs, with roles linked to the oncogenic process [19]. Although several studies have described specific miRNA expression signatures for various cancers including BL, none have described how these signatures may differ in the presence of HIV. While B-cells are not host to the virus, there is strong evidence that HIV binds to B lymphocytes through interactions with cell surface receptors [20]. HIV could thus potentially modulate miRNA expression through alteration of signalling pathways, contributing to oncogenesis. Identifying these interactions is valuable since a growing number of reports suggest that miRNAs can be useful biomarkers for pathogenic conditions, and hold therapeutic potential as they are targetable [21].

In the current study, 32 miRNAs were identified to be significantly deregulated in Burkitt lymphoma cells exposed to HIV-1 virions, compared to control cells, using a custom designed miRNA microarray. miRNA target prediction and functional annotation studies identified a majority of these miRNAs and their targets to be associated with biological processes and pathways associated with cancer. Of particular interest was the downregulation of hsa-miR-200c-3p, a miRNA that has strong associations with cancer cell invasion and migration [22,23]. Indeed, BL cells exposed to HIV-1 displayed enhanced migratory abilities. Furthermore, protein expressions of the ZEB1 and ZEB2 transcription factors, which are targets of hsa-miR-200c-3p and play key roles in cancer cell migration and invasion, were significantly enhanced in these cells. This study therefore provides evidence to support a role for HIV-1 in modulating the expression of cellular miRNAs in Burkitt lymphoma patients who are HIV positive, and that this contributes towards the cancer phenotype.

## 2. Materials and Methods

### 2.1. Cell Culture and Treatments

Burkitt lymphoma cell lines Ramos and BL41 were cultured in RPMI 1640 medium (Sigma-Aldrich, Saint Louis, MO, USA) supplemented with 10% FBS and 1% Penicillin/streptomycin (P/S). The cells were maintained at 37 °C in a humidified incubator supplemented with 5% CO_2_. Cells were exposed extracellularly with aldrithiol-2-inactivated HIV-1 virions (HIV-1 AT-2) at 500 ng/mL or matched microvesicle (MV) control, for 3 h (virions and MV controls were a kind donation of Professor Jeff Lifson, AIDS and Cancer Virus Program, Frederick National Laboratory USA) [24].

### 2.2. miRNA Isolation and PCR Array

miRNA was isolated using the *mir*Vana^TM^ miRNA Isolation kit (Thermo Fisher Scientific^TM^, Waltham, MA, USA) and quantified using the NanoDrop ND-1000 Spectrophotometer (Thermo Fisher Scientific^TM^, Waltham, MA, USA) and the Qubit^TM^ RNA Assay kit (Invitrogen, Waltham, MA, USA) as per the manufacturer’s protocol. Nucleotide integrity was analysed using gel electrophoresis. miRNA profiling was performed using an Applied Biosystems^TM^, custom 192a TaqMan^®^ Quantitative real-time PCR low density array (TLDA) card (#4346802). The 192a-card format was used, and each array card contained mature sequences of 188 miRNAs and three controls (RNU6B, RNU48, RNU44) pre-spotted in duplicate on a 384-well plate array. The RNA isolated from HIV-1 AT-2 and microvesicle treated cells (1 µg per sample) was converted to cDNA using a custom pool of multiplex stem-loop primers and the TaqMan^®^ miRNA Reverse Transcription kit (Thermo Fisher Scientific^TM^, Waltham, MA, USA), according to the manufacturer’s instructions. The cDNA samples were loaded onto the custom TaqMan^®^ Quantitative real-time PCR low density array (TLDA) and target amplification was performed using the TaqMan^®^ Universal PCR Master Mix kit (No AmpErase^®^, UNG 2X) (Thermo Fisher Scientific^TM^, Waltham, MA, USA) using specific primers and probes on the TaqMan^®^ miRNA microarray. PCR array and data analysis were performed at the Centre for Proteomic and Genomic Research (CPGR, Cape Town, South Africa) using their Applied Biosystems^TM^ 7900HT Real-Time PCR system (Applied Biosystems^TM^, Carlsbad, CA, USA).

### 2.3. miRNA Target and Pathway Analyses

The bioinformatic predictive tools TargetScan [25], DIANA TarBase [26] and miRDB [27,28] were used to identify gene targets. A list of the top 20 predicted gene targets from each bioinformatic tool was compiled for each miRNA. The Venn diagram creation tool InteractiVenn [29] was used to develop Venn diagrams for the two sets of differentially expressed miRNAs and to identify common gene targets. To identify relevant biological processes and pathways downstream of miRNA gene targets, the database for annotation, visualisation and integrated discovery (DAVID) bioinformatic tool was used [30]. Annotation of enriched biological processes and KEGG pathways downstream of target genes was restricted to those with *p* values of ≤0.05.

### 2.4. RNA Isolation and miRNA Single-Tube TAQMAN^®^ qPCR Assays

To validate differentially expressed miR-200c-3p, single-tube TaqMan^®^ miRNA assays for hsa-miR-200c-3p and endogenous controls (RNU48, RNU6B) (Applied Biosystems^TM^, Carlsbad, CA, USA) were performed. Total RNA was isolated from treated cells and reverse transcription performed (10 ng per sample) using specific stem-loop primers and the TaqMan^®^ miRNA Reverse Transcription kit (Applied Biosystems^TM^, Carlsbad, CA, USA). This was followed by qPCR using specific primer pairs (hsa-miR-200c-3p and controls) and the TaqMan^®^ Universal PCR Master Mix kit (No AmpErase^®^, UNG 2X) (Applied Biosystems^TM^, Carlsbad, CA, USA). The qPCR and analysis were performed using the Roto-Gene Q 2356 (Qiagen, Hilden, Germany). The delta CT method was used to analyse the expression of the genes of interest relative to the internal control in each of the samples. Comparison was made between the HIV-treated and control MV-treated cells using the fold-change(2^−ΔΔCT^), where the control group was set to 1.

### 2.5. cDNA Synthesis and qPCR for ZEB1 and ZEB2

Reverse transcription was performed using the iScript^TM^ cDNA Synthesis Kit (Bio Rad, Hercules, CA, USA) according to the manufacturer’s recommendations and the cDNA was used as a template for quantitative PCR using the KAPA SYBR^®^ FAST qPCR Kit (Kapa Biosystems, Western Cape, South Africa). The primer sets used for amplification were GAPDH, forward 5′-GAAGGCTGGGGCTCATTT-3′, reverse—5′-CAGGAGGCATTGCTGATGAT-3′; ZEB1, forward 5′-GCCTGAAATCCTCTCTGAATG-3′, reverse 5′-CACCTCTTGTCAAAC-3′; ZEB2, forward 5′-GAAGAGACTGGAGATCACTC-3′ and reverse 5′-GCCATCTTCCATATTGTC-3′. The expression of GAPDH was used as the internal control. Comparison was made between the HIV-treated and control MV-treated cells using the fold-change(2^−ΔΔCT^), where the control group was set to 1.

### 2.6. Total Protein Isolation and Western Blot Analyses

Total protein was isolated from treated cells using RIPA buffer (150 mM NaCl, Tris pH 7.5, 1% Triton X-100, 0.1% SDS, 10 Mm, 1% deoxycholate powder and 1 × protease inhibitor) and incubated overnight at −80 °C for optimum cell lysis. The lysed cells were centrifuged at 4 °C for 20 min and the supernatant (containing protein) was quantified using the Pierce^TM^ BCA assay kit (Thermo Fisher Scientific^TM^, Waltham, MA, USA). Twenty micrograms of protein were separated using 8% SDS-PAGE gels. The separated proteins were transferred to nitrocellulose membranes (Bio-Rad, Hercules, CA, USA) using the Mini-PROTEAN 3 casting apparatus (Bio-Rad, Hercules, CA, USA). The membrane was blocked in 5% (*w*/*v*) fat-free milk in 1 × PBS-Tween and incubated overnight at 4 °C with the following antibodies: ZEB1 (sc-25388, Santa Cruz Biotechnology, Dallas, TX, USA; 1:1000), ZEB2 (SIP1, sc-48789, Santa Cruz Biotechnology, Dallas, TX, USA; 1:1000) and p38 (Bio-Rad, Hercules, CA, USA; 1:5000). The secondary antibodies used were Goat Anti Rabbit (H + L) HRP conjugate (170-6515, Bio-Rad, Hercules, CA, USA; 1:5000) and Goat Anti Mouse (H + L) HRP conjugate (170-6516, Bio-Rad, Hercules, CA, USA; 1:5000).

### 2.7. Transwell Migration Assay

Cell migration was measured using a migration assay 2-chamber system (Transwell^®^ migration assays, Corning, NY, USA). Briefly, medium supplemented with 10% FBS was added to the bottom chamber and the Transwell^®^ chambers (8 μm pore size) were placed on top, into which cells were seeded in low-serum medium (0.5% FBS). Migration was allowed to proceed for 24 h. The cells in the upper side of the chamber were carefully removed. The migrated cells on the bottom of the membrane were fixed using 100% methanol and stained using 0.2% crystal violet, air-dried and thereafter solubilized in 50% acetic acid. The absorbance was read at 595 nm.

### 2.8. Statistical Analyses

For the miRNA PCR array data, the SDS output file (output format from the Applied Biosystem’s qRT-PCR instrument ABI7900HT) was converted to plain text using Applied Biosystem’s RQ Manager (version 1.2). Bioconductor’s HTqPCR package (Dvinge & Bertone, 2009) was used in R (R Development Core Team, 2013) to analyse the qRT-PCR data [31]. Each amplification plot was viewed using RQ Manager (Applied Biosystems^TM^, Carlsbad, CA, USA) whereby the baseline and threshold values were set manually; failed replicates were excluded and only probes with two or more replicates were retained. The data were then exported into the DataAssist^TM^ software (version 3.01) (Applied Biosystems^TM^, Carlsbad, CA, USA) to generate the Ct values for each replicate. Ct values between 30 and 37 were retained and the median value was calculated. The data were normalised using the geometric mean method [32]. The delta Ct method (2^−ΔΔCt^) was used to determine the fold change in expression of miRNAs and those that exhibited a fold change of two or more (FDR adjusted *p*-value ≤ 0.06) were selected for further analysis. Student’s *t*-test (two-tailed) was used to test for significance between the HIV and MV-treated samples. For miRNA validation and all other qPCR data, the RotorGene Q software was used to analyse and determine the Ct values. Student’s *t*-test (two-tailed) was used for comparison of the normalised data between the HIV-treated and MV-treated groups. All normally distributed data are presented as means ± SEM and significance determined using the two-sample *t*-test (Microsoft Excel for Office 365 or GraphPad Prism version 8). The latter was applied to the single-tube miRNA qPCR assays, qPCR assays for ZEB1 and ZEB2, and the migration assays.

## 3. Results

### 3.1. Exposure to HIV-1 Leads to Significant Changes in the miRNA Profile of Burkitt Lymphoma Cells

We designed a custom microarray based on the most common miRNAs reported to be deregulated in diffuse large B-cell lymphoma and Burkitt lymphoma (Appendix A). Among people living with HIV, these cancers represent the two most prevalent non-Hodgkin lymphomas (NHLs) within this population group. However, the deregulation of these miRNAs within the context of HIV remains unknown. We thus performed a differential screening, using this custom-designed array to assess changes in the expression landscape of these miRNAs within an HIV-positive context. Cells derived from the Burkitt lymphoma cell line Ramos were exposed to HIV-1 AT-2, and the miRNAome was assessed and compared to control cells (exposed to matched microvesicles). HIV-1 virions treated with Alrithiol-2 (AT-2), a mild oxidising agent, lose the ability to infect cells as a result of loss of sulphide bonds between the cysteine residues of the nucleocapsid proteins [24], but the structural integrity of the glycoproteins on the surface of the virions remains unaffected, ensuring that they retain the ability to interact with cell surface receptors. Following exposure for 3 h, thirty-two (32) miRNAs were found to be differentially expressed, by 2-fold or more (with a threshold of significance of *p* ≤ 0.06), in HIV exposed cells when compared to control cells (Figure 1a). This therefore indicates that the pathobiology of HIV-associated NHLs provides a cellular microenvironment that alters miRNA pathways in a way that is distinct from one where HIV is not present.

We next sought to better define the role of these miRNAs in order to understand how they contribute to the HIV-associated NHL cancer phenotype. Using three independent predictive bioinformatics tools (TargetScan [25], DIANA TarBase [26] and miRDB [27,28]), the top 20 genes potentially targeted by each of the 32 miRNAs were identified and analysed, revealing 13 and 23 genes commonly targeted among all three databases, for upregulated and downregulated miRNAs, respectively (Figure 1b and Table 1). The analysis of biological processes and pathways associated with these 36 genes reveals associations with a variety of cellular processes and pathways, but notably with B-cell differentiation, cell cycling, proliferation, DNA damage and drug responses, and several typical KEGG cancer pathways **(**Figure 2). For instance, miR-222-3p, which we found to be upregulated by ~7-fold in our array, has previously been shown to be downregulated in BL relative to DLBCL [33], and specifically targets the cyclin-dependent kinase inhibitor p27^Kip1^ [34]. Within an HIV-positive environment, an upregulation of this miRNA would therefore translate to enhanced cellular proliferation, a pertinent feature of these aggressive HIV-associated cancers. Other miRNAs with known roles in cancer-promoting processes identified in the array include miR-575, miR-363-3p and several others. Of particular interest was hsa-miRNA-200c-3p, which, in addition to having a high association with numerous cancer types, has previously been reported to be downregulated in DLBCL [35].

### 3.2. Hsa-miR-200c-3p Is Significantly Downregulated in HIV-1 Treated Burkitt Lymphoma Cells, and This Is Associated with Enhanced Migration

We found the expression of hsa-miR-200c-3p to be downregulated by greater than 6.67-fold in the array. The validity of this observation was strengthened when single-tube miRNA assays showed that miR-200c-3p was indeed significantly downregulated in the Ramos cells (2-fold), as well as in a second BL cell line, BL41 (2-fold), when these cells were exposed to HIV-1, relative to controls (Figure 3a,b). The miRNA-200 family, which is highly conserved among vertebrates, has been shown to play a key role in cancer, from cancer initiation to metastasis [22]. Not only has it been shown to be downregulated in B-cell lymphomas, but also in cancers of the breast, lung, oesophagus, stomach, colon and many others. Importantly, the use of this miRNA as a prognostic marker looks promising, showing a favourable positive predictive value when evaluated in the plasma levels of cancer patients. Although reported to be involved in a variety of cancer types and cellular processes (Table 2), the miRNA-200 family is particularly associated with inhibition of the epithelial-to-mesenchymal transition, an early step in metastasis, by maintaining the epithelial phenotype through directly targeting the transcriptional repressors [36]. In lymphoma, the role of miRNA-200c remains unclear, with reports of both upregulation and downregulation of this microRNA [37,38]. In order to ascertain whether this downregulation was physiologically relevant, the migratory ability of the BL cells when exposed to HIV-1 was investigated. A validated in vitro assay was used, consisting of a two-chamber system separated by a porous membrane, and differential chemoattractant (FBS) in the two chambers. The extent of cellular migration was determined 24 h post-treatment. Indeed, both Ramos and BL41 cells displayed significantly enhanced migratory abilities in the presence of HIV-1, compared to control cells. The migration rates increased by 32% and 37% for Ramos and BL41 cells, respectively (Figure 3c,d).

### 3.3. MiR-200c-3p Downregulation and Enhanced Migration Correlates with Over-Expression of ZEB1 and ZEB2 Proteins in BL Cells

The Zinc Finger E-box Binding (ZEB) family of transcription factors has been experimentally confirmed in numerous studies to be targeted by miR-200c [39,49]. These proteins have been described as master regulators of epithelial-to-mesenchymal transition (EMT), through their ability to regulate genes involved in cell plasticity, intercellular adhesions and degradation of the extracellular matrix [50]. ZEB1 was one of the top 20 targets, from three databases, found to be potentially upregulated in our array (Table 1). At the mRNA level, we found ZEB1 to be significantly downregulated in both Ramos (1.75-fold) and BL41 cells (1.30-fold), when exposed to HIV-1 (Figure 4a,b). Conversely, at the protein level, there was an increase in ZEB1 protein expression in both the Ramos (2.75-fold) and BL41 (1.33-fold) cells (Figure 4c,d). A very similar pattern was observed when the expression of ZEB2 was investigated. The expression of the ZEB2 mRNA was significantly reduced in both cell lines upon exposure to HIV-1 (Figure 5a,b), with a decrease of 2.04-fold in Ramos cells, and of 1.30-fold in BL41 cells. As for ZEB1, the expression of the ZEB2 protein was enhanced in both the Ramos cells (1.99-fold), as well as the BL41 cells (2.78-fold) upon exposure to HIV-1 (Figure 5c,d).

## 4. Discussion

People living with HIV are at increased risk of developing cancer, with non-Hodgkin lymphoma being one of the most prevalent cancers within this group [8]. While traditionally this enhanced risk was associated with HIV-1-induced immune suppression and exhaustion, as well as chronic B-cell activation, the advent of antiretroviral therapy (ART), even at early stages of infection, did not abolish this risk [51]. There is now enough experimental evidence to support an oncogenic role for HIV-1 and its antigens in carcinogenesis [52]. HIV-1 does not infect B lymphocytes; however, the virus is capable of binding these cells through cell surface receptors [20], and so can components of the virus, as has been demonstrated by binding of the p17 matrix protein to the CXCR receptors [53].

Whether through cell surface signalling, or via internalization, HIV-1 has the ability to alter cellular processes at multiple levels. In the current study, using a custom array design based on frequently reported altered miRNAs in the two most prevalent HIV-associated NHLs, we identified 32 miRNAs that were differentially expressed, out of 188 selected, in BL cells exposed to HIV-1, relative to controls. To the best of our knowledge, this is the first study to report on differentially expressed miRNAs in Burkitt lymphoma cells exposed to HIV-1. The relationship between HIV-1 and cellular miRNAs is well described. The virus and components of the virus have previously been reported to alter expression of cellular miRNAs in other cellular contexts. In a recent study, the T-cell lymphoblastic lymphoma SupT1 cell line showed alteration of several cellular miRNAs upon infection with HIV [54]. In fact, the alteration of host miRNA networks in CD4+ cells seems to be crucial for successful viral invasion and latency. Within the context of HIV non-host cells, there are a few reports of alterations in miRNAs due to the presence of the virus or its antigens. For instance, in Kaposi’s sarcoma (KS), the HIV-1 Tat protein has been shown to synergize with the KSHV oncogene Orf-K1 to induce miR-891a-5p, modulating NF-kB [55]. There has not, as yet, been any comprehensive study on large-/medium-scale differential miRNA expression in NHLs comparing HIV-positive with HIV-negative. An earlier study conducted on a cohort in Kenya measured the expression of only a selected number of miRNAs, linked to the regulation of DNA methyltransferase (DNMT), from HIV-related NHLS (formalin-fixed paraffin embedded tumours) and compared that to expression in HIV-negative controls [56].

MiRNA in silico prediction analyses allowed for the identification of 36 putative targets (13 potentially downregulated, and 23 potentially upregulated). The miRNA interactome is complex, with single miRNAs shown to be able to target dozens of genes, and this therefore hinders straightforward interpretation of differences in miRNA expression. Importantly, a majority of the predicted targets identified from our miRNA array, and their associated biological processes, were found to be associated with cancer hallmarks with a high degree of confidence. Nevertheless, experimental validation is essential to assess true physiological impact, and thus, using single-tube miRNA assays, hsa-miR-200c-3p was confirmed as significantly downregulated in BL cells exposed to HIV-1. This downregulation was strongly associated with enhanced cellular migration, a physiological function linked to this miRNA (36). The role and significance of miR-200c-3p in BL development has not been clearly defined. In many cancers such as breast, ovarian and endometrial cancers, miR-200c-3p has been identified to have a tumour suppressor role [41,48,57]. The miR-200 family cluster of miRNAs have been termed “the guardians of the epithelial phenotype”, as they have been found to be enriched in epithelial tissues and epigenetically silenced in mesenchymal tissues [42,58]. Along with other miRNA, members of the miR-200 family have been shown to be a marker of the epithelial phenotype since miR-200c-3p targets numerous mesenchymal genes and inhibits tumour cell migration and invasion [59].

ZEB1 and ZEB2 are two mesenchymal genes that are direct targets of miR-200c-3p [36,39,48,60] and we thus sought to find a correlation between them within our model system. We found the mRNA expression of both ZEB1 and ZEB2 to be significantly downregulated in response to HIV-1 exposure in both cell lines (Figure 4a,b and Figure 5a,b). Although an inverse correlation is expected, assessment of changes in the transcription level of miRNA targets can be misleading. This is because miRNA may bind to mRNA and prevent translation, with no change in mRNA levels, and this has been demonstrated in several studies [61]. We speculate that the downregulation of ZEB transcription observed upon HIV-1 exposure in our study results from a different mechanism and could represent an attempt by the cellular machinery to mitigate the deleterious effects of this highly oncogenic factor. ZEB is known to be regulated at multiple levels through a complex web of intracellular signalling pathways and it would be difficult to pinpoint the exact mechanism without a global view of the regulatory landscape within B cells in an HIV-positive background, and this supports the need for more research in this field [62]. It is, however, important to note that ZEB transcription is not abrogated when exposed to HIV-1, and sufficient mRNA is still being produced to lead to enhanced protein expression.

Interestingly, there is a double-negative feedback loop that exists between the ZEB transcription factors and the miR-200 family, where these proteins can directly bind to and inhibit the expression genes encoding for the miR-200 family [63,64]. This feedback loop is advantageous to cells, as it allows for easy and reversible switching between epithelial and mesenchymal characteristics, depending on extracellular signals [65]. Contrary to the mRNA, the expression of ZEB1 and ZEB2 protein was upregulated in Burkitt lymphoma cells upon exposure to HIV-1, supporting the notion that the enhanced migration of BL cells via downregulation of hsa-miR-200c-3p leads to an alleviation of active repression of the ZEB proteins.

Although still poorly understood, it is important to note that the mechanisms driving malignant cell migration are highly complex and involve several coordinated events including development of cytoplasmic protrusions, changes in cellular adhesion and traction, expression of proteolytic enzymes and more [66]. The ZEB proteins are part of a group of transcription factors (including Snail, Slug, KLF8 and others) that regulate this process, and while they have been shown to be specifically involved with cell plasticity, for instance through their ability to repress the adhesion molecule E-cadherin, they contribute to several other cellular events that promote cancer, such as enhancing cell cycling and acquisition of drug resistance [64]. Similarly, although the role of hsa-miR-200c-3p in cancer cell migration is well described, several other miRNAs have been identified to play pertinent roles in this specific function [67,68]. It is therefore clear that an array of factors and mechanisms is needed to drive cancer cell migration, and that in this particular study, while the hsa-miR-200c-3p/ZEB axis has been identified as a role player, it is as yet an association that needs to be confirmed via further loss- and gain-of-function analyses.

This study therefore contributes to accumulating evidence that HIV-1 can directly promote oncogenic pathways in B-cell lymphoma, and clinical studies should be conducted to evaluate the use of hsa-miR-220c-3p as a potential novel biomarker that can be used for prognosis in patients with Burkitt lymphoma, who are HIV positive.

## Figures and Tables

**Figure 1 genes-12-01302-f001:**
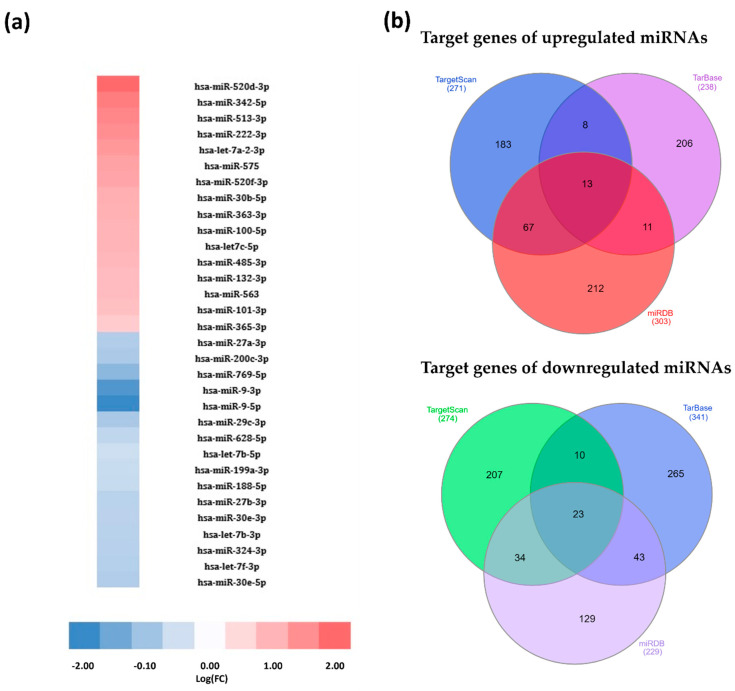
miRNAs differentially expressed by 2-fold or more (with a threshold of significance of *p* ≤ 0.06) in Ramos cells exposed to HIV-1 AT-2 compared to matched microvesicle cells (control), and alteration of predicted genes. (**a**) Heat map showing differentially expressed miRNAs, data represented as log(FC). (**b**) Venn diagram showing predicted miRNA gene targets identified by three independent bioinformatic tools (TargetScan, Tarbase and miRDB).

**Figure 2 genes-12-01302-f002:**
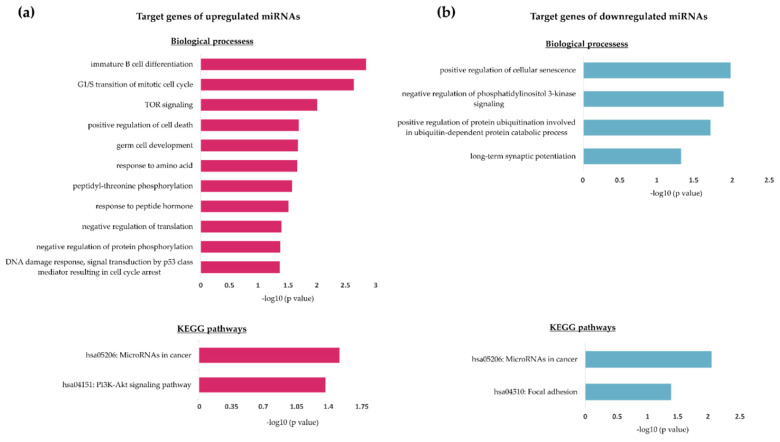
Biological process and KEGG pathway analysis of predicted target genes of differentially expressed miRNAs in Ramos cells exposed to HIV-1 AT-2. Biological processes (upper bar graph) and KEGG pathways (lower bar graph) significantly associated with predicted gene targets of upregulated (**a**) and downregulated (**b**) miRNAs. Data are represented as −log10 (*p* value); only significantly enriched associations were annotated (*p* < 0.05).

**Figure 3 genes-12-01302-f003:**
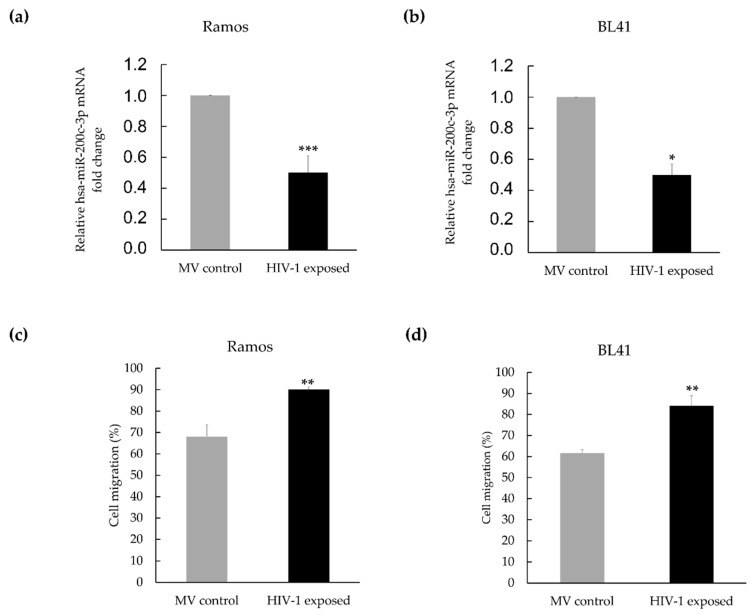
The downregulation of hsa-miR-200c-3p in HIV-1 AT-2 treated cells correlates with enhanced cell migration. (**a**) Fold change in miR-200c-3p expression in Ramos and (**b**) in BL41 cells exposed to HIV-1 AT-2 compared to control microvesicle-exposed cells. The cells were treated with either HIV-1 AT-2 or microvesicles and thereafter RNA was isolated. TaqMan^®^ single-tube miRNA assays were used for RT-qPCR and the delta Ct (2^−ΔΔCt^) method was used for quantification. (**c**) Fold change in migration in Ramos cells and (**d**) in BL41 cells exposed to HIV-1 AT-2 compared to control microvesicle-exposed cells. Cells were treated as described above and Transwell^®^ migration assays were used to measure migratory ability. The treated cells were plated in low-serum medium (0.5% FBS) on the top chamber, and allowed to migrate to the bottom nutrient-rich (10% FBS) medium. Migrated cells were stained and absorbance readings (correlating to the number of cells) were taken. The data were normalised to the total number of plated cells. Student’s *t*-test was performed to determine statistical significance. (* *p* < 0.05, ** *p* < 0.01, *** *p* < 0.001) and error bars represent standard deviation.

**Figure 4 genes-12-01302-f004:**
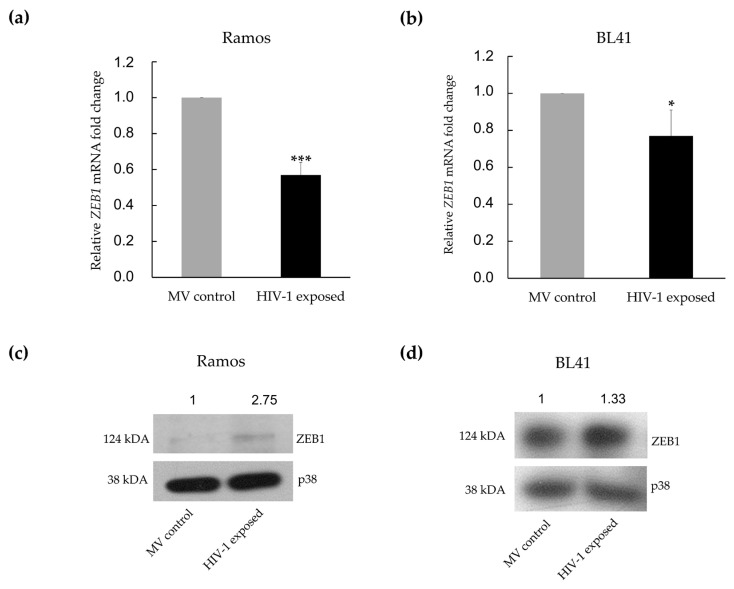
HIV-1 AT-2 deregulates expression of ZEB1 in BL cells. The cells were treated with either HIV-1 AT-2 or microvesicles and thereafter RNA (**a**,**b**) or protein (**c**,**d**) was isolated. mRNA expression of ZEB1 in Ramos (**a**) and BL41 (**b**) cells treated with HIV-1 AT-2 as determined by RT-qPCR. Protein expression of ZEB1 in Ramos (**c**) and BL41 (**d**) cells, as determined by Western blotting, using p38 as loading control. For (**a**,**b**), the delta Ct (2^−ΔΔCt^) method was used and Student’s *t*-test was performed to determine statistical significance. (* *p* < 0.05, *** *p* < 0.001) and the error bars represent standard deviation.

**Figure 5 genes-12-01302-f005:**
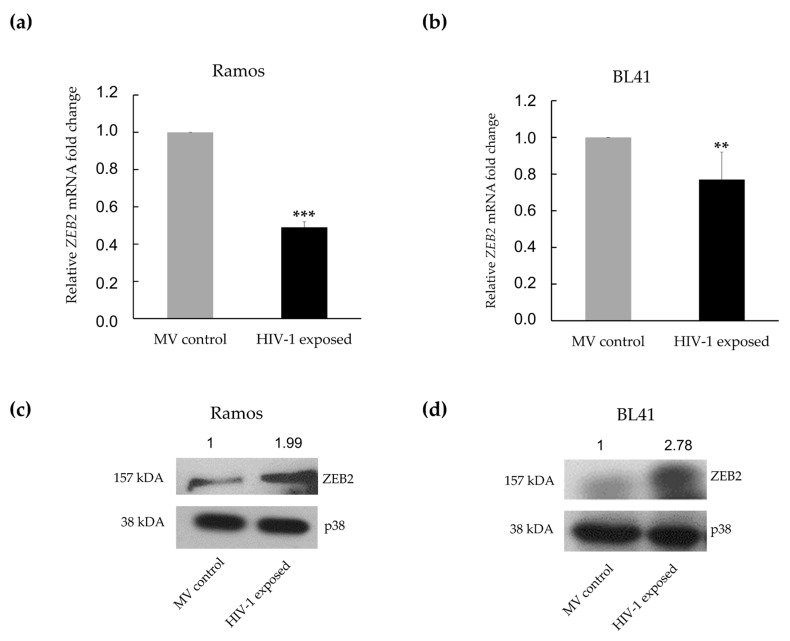
HIV-1 AT-2 deregulates expression of ZEB2. The cells were treated with either HIV-1 AT-2 or microvesicles and thereafter RNA was isolated after treatment. mRNA expression of ZEB2 in (**a**) Ramos and (**b**) BL41 cells treated with HIV-1 AT-2 was determined using RT-qPCR. Western blot and quantification analysis of ZEB2 protein expression in (**c**) Ramos and (**d**) BL41 cells treated with HIV-1 AT-2 compared to control cells. For (**a**,**b**), the delta Ct (2^−ΔΔCt^) method was used for quantification and Student’s *t*-test was performed to determine statistical significance. (** *p* < 0.01, *** *p* < 0.001) and the error bars represent standard deviation.

**Table 1 genes-12-01302-t001:** List of commonly predicted gene targets of upregulated and downregulated miRNAs in BL cells exposed to HIV-1.

Predicted Gene Targets of Upregulated miRNAs (13)	Predicted Gene Targets of Downregulated miRNAs (23)
*CDKN1B*, *KIT*, *MIDN*, *ANKRA2*, *FNIP1*, *BTG2*, *HS3ST2*, *CTDSPL*, *SMARCA5*, *MTOR*, *TRIM71*, *MYCN*, *ACVR1*	*NOVA1*, *FBXW7*, *ZEB1*, *GABRA1*, *POU2F1*, *MTHFD2*, *LIN28B*, *TET1*, *COL3A1*, *HMGA2*, *KLHL3*, *SERPINE2*, *PAWR*, *GNA12*, *PAK4*, *BCAR3*, *CD2AP*, *FOXN2*, *ZFP91*, *SLC22A3*, *PTEN*, *C6orf106*, *SRSF7*

**Table 2 genes-12-01302-t002:** Selected list of experimentally validated gene targets of hsa-miR-200c-3p and the supporting literature.

Gene Target	Validation Tool	Cancer Type: Process	Reference
Zinc Finger E-box Binding Protein 1 and 2 (ZEB1, ZEB2)	Dual-luciferase reporter assay, Western blotting, qPCR	Gastric, Head and neck squamous cell carcinoma (HNSCC): Epithelial-to-mesenchymal transition (EMT), migration and invasion.	[23,39]
X-linked inhibitor of apoptosis protein (XIAP), B-cell lymphoma 2 (BCL2)	Dual-luciferase reporter assay, Western blotting	Gastric, Lung: apoptosis	[40]
Tubulin β 3 Class III (TUBB3)	Dual-luciferase reporter assay, Western blotting, qPCR	Breast, Ovarian, endometrial: proliferation, drug resistance	[41,42]
KRAS proto-oncogene (KRAS)	Dual-luciferase reporter assay, Western blotting	Breast, NSCLC: Proliferation	[43]
B-cell-specific Moloney murine leukaemia virus insertion site 1 (BMI1), E2F transcription factor 3 (E2F3)	Reporter assay, Western blotting, qPCR	Renal, Bladder: Proliferation, migration, and invasion	[44,45]
Fibronectin 1 (FN1), Moesin (MSN)	Reporter assay, Western blotting, qPCR	Breast, endometrial: Invasion, anoikis resistance	[46,47]
Inhibitor of Nuclear Factor Kappa β Kinase Subunit β (IKBKβ), Fms Related Receptor Tyrosine Kinase 1 (FLT1), Kruppel Like Factor 9 (KLF9)	Reporter assay, Western blotting, qPCR	Endometrial: Proliferation, inflammation	[48]

## Data Availability

Not applicable.

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
