# Peer review of "Modulation of Cellular MicroRNA by HIV-1 in Burkitt Lymphoma Cells—A Pathway to Promoting Oncogenesis"

_genes, 2021, doi:10.3390/genes12091302_

Round 1

Reviewer 1 Report

Ramorola and colleagues presented an interesting work demonstrating the ability of HIV-1 to induce in Burkitt lymphoma cells a different expression of miRNAs mainly associated with oncogenic processes. In general, this work can represent an advance in the knowledge of novel mechanisms underlying the establishment of a relationship between HIV-1 and host gene expression, particularly focusing on the induction of microRNAs dysregulation in B-cells in Burkitt lymphoma.

The manuscript is very well written and theorically well designed, however I have the following major concerns regarding this work.

  1. In the Abstract, the following phrase should be scaled down “… and identifies hsa-miR-200c-3p as a novel biomarker 22 for prognosis in patients with Burkitt lymphoma, who are HIV positive.” because in the study there is not the evaluation of has-miR-200c-3p as prognostic marker in patients with Burkitt lymphoma, who are HIV positive.
  2. In the Results:
  • Line 193, “Using three independent predictive bioinformatics tools…”, please specify which tools.
  • Figure 1: please, indicate the statistical threshold used.
  • Figure 2 is cut and comes out of the sheet
  • Figure 3: in the figure legend the following phrase should be scaled down “The downregulation of has-miR-200c-3p in HIV-1 AT-2 treated cells leads to enhanced cell migration”, because the results obtained in this study can not allow to evaluate a direct functional modulation of has-miR-200c-3p on cell migration in B cells infected by HIV-1.
  1. In Materials and Methods:
  • What amount of microRNA was used in both the PCR array and the miRNA single-tube Taqman qPCR assays?
  • At the end of section 2.2, PCR array and data analysis deserve a more exaustive description by indicating methods, tools, normalization creteria, eventual statistical correction, etc. This information is partly reported at the beginning of section 2.8, but is not complete, for example statistical test, method of normalization, statistical significance theshold, and other parameters are not described.
  • In the section 2.8, only the Student’s t-test (two-tailed) was indicated and it is unclear for which dataset it was applied. The authors should report the statistical test used to analyze each individual type of experiment.
  • There is no information on how long after HIV-1 infection the RNA and proteins were extracted.

Author Response

Comments and Suggestions for Authors

Ramorola and colleagues presented an interesting work demonstrating the ability of HIV-1 to induce in Burkitt lymphoma cells a different expression of miRNAs mainly associated with oncogenic processes. In general, this work can represent an advance in the knowledge of novel mechanisms underlying the establishment of a relationship between HIV-1 and host gene expression, particularly focusing on the induction of microRNAs dysregulation in B-cells in Burkitt lymphoma.

The manuscript is very well written and theorically well designed, however I have the following major concerns regarding this work.

  1. In the Abstract, the following phrase should be scaled down “… and identifies hsa-miR-200c-3p as a novel biomarker 22 for prognosis in patients with Burkitt lymphoma, who are HIV positive.” because in the study there is not the evaluation of has-miR-200c-3p as prognostic marker in patients with Burkitt lymphoma, who are HIV positive.

Author response:

The above point is duly recognised. We have changed the sentence “This study therefore identifies novel miRNAs as role players in the development of HIV-associated BL, and identifies hsa-miR-200c-3p as a novel biomarker for prognosis in patients with Burkitt lymphoma, who are HIV positive.” to “This study therefore identifies novel miRNAs as role players in the development of HIV-associated BL, with one of these miRNAs, hsa-miR-200c-3p, being a candidate for further clinical studies as a potential biomarker for prognosis in patients with Burkitt lymphoma, who are HIV positive.” – Lines 21 - 24

  1. In the Results:
  • Line 193, “Using three independent predictive bioinformatics tools…”, please specify which tools.

Author response: The information (which is displayed in the Materials and Methods section of the original manuscript), has been inserted in this section as well (line 212).

  • Figure 1: please, indicate the statistical threshold used.

Author response: The threshold of significance has been included (Lines 205-207): Following exposure for 3 hours, thirty two (32) miRNAs were found to be differentially expressed, by 2-folds or more (with a threshold of significance of p ≤ 0.06), in HIV exposed cells when compared to control cells (Figure 1a).

  • Figure 2 is cut and comes out of the sheet

Author response: We apologize for this oversight. The size of the Figure has been adjusted to fit within the page.

  • Figure 3: in the figure legend the following phrase should be scaled down “The downregulation of has-miR-200c-3p in HIV-1 AT-2 treated cells leads to enhanced cell migration”, because the results obtained in this study can not allow to evaluate a direct functional modulation of has-miR-200c-3p on cell migration in B cells infected by HIV-1.

Author response: This point is duly noted. The sentence “The downregulation of has-miR-200c-3p in HIV-1 AT-2 treated cells leads to enhanced cell migration“ has been changed to “The downregulation of has-miR-200c-3p in HIV-1 AT-2 treated cells correlates with enhanced cell migration” – Lines 272-273.

  1. In Materials and Methods:
  • What amount of microRNA was used in both the PCR array and the miRNA single-tube Taqman qPCR assays?

Author response: An amount of 1µg per sample was used for the PCR array (this information has been inserted in the materials and methods section – line 99). An amount of 10ng per sample was used for the single-tube assays (this information has been inserted in the materials and methods section – line 123).

  • At the end of section 2.2, PCR array and data analysis deserve a more exaustive description by indicating methods, tools, normalization creteria, eventual statistical correction, etc. This information is partly reported at the beginning of section 2.8, but is not complete, for example statistical test, method of normalization, statistical significance theshold, and other parameters are not described.

Author response: We have now substantially revised the narrative in section 2.8, and inserted a detailed description of the statistical analyses employed for the PCR array, as well as including the statistically analyses which were used for the other assays used in the study (Lines 168 – 188).

  • In the section 2.8, only the Student’s t-test (two-tailed) was indicated and it is unclear for which dataset it was applied. The authors should report the statistical test used to analyze each individual type of experiment.

Author response: The type of statistical test used, and the experiment those were applied to, have been included in section 2.8.

  • There is no information on how long after HIV-1 infection the RNA and proteins were extracted.

Author response: The length of treatment is 3 hours, and can be found in the material and methods section (section 2.1). For further clarity, this information has been included under the Results section as well, at line 205.

Reviewer 2 Report

The functions of microRNAs in the pathobiology of HIV-associated cancers are unclear. In this study, the authors performed a miRNA screening PCR to identify differently expressed miRNAs in Burkitt lymphoma cells exposed to HIV-1. Has-miR-200c-3p was found to play a crucial role in cancer cell migration. In addition, the ZEB1 and ZEB2 transcription factors were confirmed to be direct targets of has-miR-200c-3p. However, the creativity of this manuscript is insufficient. Majority of the results are descriptive summary reports from publicly available datasets and not very logical and rational.  

Specific comments to further improve the manuscript are as follows:

  1. The deregulated miRNAs have been reported in previous studies. Please highlight the authors’ original discoveries. miRNAs target genes name list should not be presented in Figure 1C, but in a table.
  2. Ramos and BL41 cells displayed significantly enhanced migratory abilities in the presence of HIV-1. However, this finding could not demonstrate the direct function of has-miR-200c-3p in cancer cell migration. To further verify the function of miR-493-5p on malignant biological properties, the authors should design the mimic and inhibitor sequences of has-miR-200c-3p to change the expression level. Moreover, the rationale to investigate has-miR-200c-3p should be well addressed, Not only hsa-miR-200c-3p was downregulated by greater than 6-8 folds in the array.
  3. The association between hsa-miR-200c-3p and ZEB1 should be verified by the gain-or loss- function studies.
  4. The authors indicated that hsa-miR-200c-3p serves as a novel biomarker for prognosis in patients with Burkitt lymphoma, who are HIV positive. However, there is no clinical data to support this conclusion.

Minor points

  1. Figure 2 information is not complete.
  2. In line 75-76, “migration in invasion” is elusive
  3. In line 246 "has-miR-200c-3p"

Author Response

Comments and Suggestions for Authors

The functions of microRNAs in the pathobiology of HIV-associated cancers are unclear. In this study, the authors performed a miRNA screening PCR to identify differently expressed miRNAs in Burkitt lymphoma cells exposed to HIV-1. Has-miR-200c-3p was found to play a crucial role in cancer cell migration. In addition, the ZEB1 and ZEB2 transcription factors were confirmed to be direct targets of has-miR-200c-3p. However, the creativity of this manuscript is insufficient. Majority of the results are descriptive summary reports from publicly available datasets and not very logical and rational.  

Specific comments to further improve the manuscript are as follows:

  1. The deregulated miRNAs have been reported in previous studies. Please highlight the authors’ original discoveries. miRNAs target genes name list should not be presented in Figure 1C, but in a table.

Author response: The custom miRNA array was designed to include miRNAs which have previously been described to be deregulated in non-Hodgkin lymphoma, but these miRNAs had not previously been described within an HIV positive context, which represents the novelty in this study. A clearer description of the research design, as well as novel discoveries, has been included (lines 192 – 196; lines 205-209; 219-223).

Thank you. We agree that the list of target genes are best represented as a Table. We have removed Figure 1C and placed the list as Table 1.

  1. Ramos and BL41 cells displayed significantly enhanced migratory abilities in the presence of HIV-1. However, this finding could not demonstrate the direct function of has-miR-200c-3p in cancer cell migration. To further verify the function of miR-493-5p on malignant biological properties, the authors should design the mimic and inhibitor sequences of has-miR-200c-3p to change the expression level. Moreover, the rationale to investigate has-miR-200c-3p should be well addressed, Not only hsa-miR-200c-3p was downregulated by greater than 6-8 folds in the array.

Author response: We agree that, in order to directly correlate the enhanced migratory abilities of HIV-exposed BL cells with the downregulation of hsa-miR-200c-3p and upregulation of ZEB proteins, we will have to conduct further assays using mimics and inhibitors of the miRNA. While these experiments are being planned for the future, we are not able, at this stage, to predict when we will have these results (the timeline prediction is even more uncertain due to the evolving COVID-19 related restrictions imposed in South Africa which has significantly impacted research). In the current study we therefore emphasize that the HIV-1-miR300c-3p-ZEB1/2 axis is a novel association which forms the basis for further future research. Furthermore, we have included a paragraph in the discussion section (lines 392 – 406) to emphasize the complexity of the factors involved in driving cellular migration, that our study shows an interesting association which will need to ne confirmed using loss- and gain-of-function studies.

Indeed, the rational for the focus on miR-200c-3p was not clear. We have elaborated upon this and the information can be found in the results section (lines 250-261).

  1. The association between hsa-miR-200c-3p and ZEB1 should be verified by the gain-or loss- function studies.

Author response: Several studies, using gain-or-loss of function experiments, have reported on the direct regulation of ZEB1 by hsa-miR-200c-3p, and these are displayed in Table 2. We agree that performing gain-or-loss of function studies within our own cells will allow us to further confirm this association and we have planned this in future studies. In the current study we report on a novel potential association of the downregulation of hsa-miR-200c-3p with enhanced migration in HIV exposed BL cells, and that this is potentially driven by enhanced expression of ZEB proteins.

  1. The authors indicated that hsa-miR-200c-3p serves as a novel biomarker for prognosis in patients with Burkitt lymphoma, who are HIV positive. However, there is no clinical data to support this conclusion.

Author response: We indeed report hsa-miR-200c-3p as a potential biomarker, based on our result, and indicate that this will have to be validated in clinical studies (lines 23 and 408).

Minor points

  1. Figure 2 information is not complete.

Author response: We apologize for this oversight. We have adjusted the Figure such that it now fits within the page.

  1. In line 75-76, “migration in invasion” is elusive

Author response: This has been corrected to read “migration and invasion”.

  1. In line 246 "has-miR-200c-3p"

Author response: Thank you, we have corrected this.

Reviewer 3 Report

This is an interesting study describing the downregulation of hsa-miR-200c-3p micro RNA in Burkitt lymphoma cells following their exposure to HIV-1 . The authors show that this downregulation is associated with increased migration of BL cells after exposure to HIV-1. In addition the expression of ZEB1 and ZEB2 transcription factors is increased at the protein level and this is associated with tumor invasion and metastasis. The authors conclude that hsa-miR-200c-3p may represent a novel prognostic biomarker for HIV positive Burkitt lymphoma patients

Comments

  1. It is not clearly shown the reason for selection of this particular hsa-miR-200c-3p micro RNA. Was it because of the highest modification  in its expression following the exposure of BL cells to HIV-1 or because of its pathogenetic role in DLBCL. The authors should more clearly define this
  2. The authors have shown that BL cells show increased migration ability following their exposure to HIV1 but there is only an indirect indication from these experiments that this increased migration capacity is associated with the reduction in the expression of hsa-miR-200c-3p since other factors may also be involved. This should be clarified
  3. Although ZEB1 and ZEB2 mRNAs are expected to increase following the exposure of BL cells to HIV-1 the opposite was found. The explanation provided by the authors is that micro RNAS bind to mRNAs and prevent translation without changes at the mRNA level. However no explanation is provided about the reduction of ZEB1 and ZEB2 mRNAs. Moreover the ZEB1 and ZEB2 proteins as expected are significantly increased in BL cells exposed to HIV-1 and the explanation provided is that enhanced migration of BL cells due to the downregulation of hsa-miR-200c-3p leads to the alleviation of active repression of ZEB proteins. However these proteins were determined in a closed culture system and the migration capacity of the cells could not be considered. The authors should more clearly explain the discrepancy between mRNA and protein ZEB1 and ZEB2 expression

4.The statement that hsa-miR-200c-3p is a novel prognostic biomarker in HIV patients with BL is rather strong and changed to “possible prognostic biomarker if verified in patients’ sample

Author Response

Comments and Suggestions for Authors

This is an interesting study describing the downregulation of hsa-miR-200c-3p micro RNA in Burkitt lymphoma cells following their exposure to HIV-1 . The authors show that this downregulation is associated with increased migration of BL cells after exposure to HIV-1. In addition the expression of ZEB1 and ZEB2 transcription factors is increased at the protein level and this is associated with tumor invasion and metastasis. The authors conclude that hsa-miR-200c-3p may represent a novel prognostic biomarker for HIV positive Burkitt lymphoma patients

Comments

  1. It is not clearly shown the reason for selection of this particular hsa-miR-200c-3p micro RNA. Was it because of the highest modification  in its expression following the exposure of BL cells to HIV-1 or because of its pathogenetic role in DLBCL. The authors should more clearly define this

Author response: During our initial validation of the miRNA PCR array result, we selected four miRNAs, of which hsa-miR-200c-3p was one. The selection of each one was based on reports in the published literature on the role and potential as biomarkers in cancer. We have now elaborated on the reason for our focus on hsa-miR-200c-3p which is included in the result section – lines 250-261.

  1. The authors have shown that BL cells show increased migration ability following their exposure to HIV1 but there is only an indirect indication from these experiments that this increased migration capacity is associated with the reduction in the expression of hsa-miR-200c-3p since other factors may also be involved. This should be clarified

Author response: This is indeed correct. A paragraph has now been included in the discussion section (Lines 392 – 406) which elaborates on the complexity of the cancer cell migration mechanism, and that further studies (loss- and gain-of-function) must be conducted to confirm a direct association.

  1. Although ZEB1 and ZEB2 mRNAs are expected to increase following the exposure of BL cells to HIV-1 the opposite was found. The explanation provided by the authors is that micro RNAS bind to mRNAs and prevent translation without changes at the mRNA level. However no explanation is provided about the reduction of ZEB1 and ZEB2 mRNAs. Moreover the ZEB1 and ZEB2 proteins as expected are significantly increased in BL cells exposed to HIV-1 and the explanation provided is that enhanced migration of BL cells due to the downregulation of hsa-miR-200c-3p leads to the alleviation of active repression of ZEB proteins. However these proteins were determined in a closed culture system and the migration capacity of the cells could not be considered. The authors should more clearly explain the discrepancy between mRNA and protein ZEB1 and ZEB2 expression

Author response: Thank you for this valid comment. While we cannot provide a definite explanation for the observed decrease in mRNA expression, we have observed this phenomenon in at least three independent biological replicates and therefore have confidence that it is a true reflection of biological events. We have elaborated on possible scenarios which could explain this, and provided an explanation in the discussion section (lines 374 – 382).  

4.The statement that hsa-miR-200c-3p is a novel prognostic biomarker in HIV patients with BL is rather strong and changed to “possible prognostic biomarker if verified in patients’ sample

Author response: We have changed this narrative in both the abstract (lines 21-24) and the discussion (lines 408-410).

Round 2

Reviewer 2 Report

Please upload the updated version using 'Track changes'. 

Author Response

Thank you for the opportunity to revise. We have tracked the changes made, as requested. Please find updated responses below (the line position of changes have been adjusted in the response):

Comments and Suggestions for Authors

The functions of microRNAs in the pathobiology of HIV-associated cancers are unclear. In this study, the authors performed a miRNA screening PCR to identify differently expressed miRNAs in Burkitt lymphoma cells exposed to HIV-1. Has-miR-200c-3p was found to play a crucial role in cancer cell migration. In addition, the ZEB1 and ZEB2 transcription factors were confirmed to be direct targets of has-miR-200c-3p. However, the creativity of this manuscript is insufficient. Majority of the results are descriptive summary reports from publicly available datasets and not very logical and rational.  

Specific comments to further improve the manuscript are as follows:

  1. The deregulated miRNAs have been reported in previous studies. Please highlight the authors’ original discoveries. miRNAs target genes name list should not be presented in Figure 1C, but in a table.

Author response: The custom miRNA array was designed to include miRNAs which have previously been described to be deregulated in non-Hodgkin lymphoma, but these miRNAs had not previously been described within an HIV positive context, which represents the novelty in this study. A clearer description of the research design, as well as novel discoveries, has been included (lines 197 – 201; lines 210-212; 225-227).

Thank you. We agree that the list of target genes are best represented as a Table. We have removed Figure 1C and placed the list as Table 1 (Line 238).

  1. Ramos and BL41 cells displayed significantly enhanced migratory abilities in the presence of HIV-1. However, this finding could not demonstrate the direct function of has-miR-200c-3p in cancer cell migration. To further verify the function of miR-493-5p on malignant biological properties, the authors should design the mimic and inhibitor sequences of has-miR-200c-3p to change the expression level. Moreover, the rationale to investigate has-miR-200c-3p should be well addressed, Not only hsa-miR-200c-3p was downregulated by greater than 6-8 folds in the array.

Author response: We agree that, in order to directly correlate the enhanced migratory abilities of HIV-exposed BL cells with the downregulation of hsa-miR-200c-3p and upregulation of ZEB proteins, we will have to conduct further assays using mimics and inhibitors of the miRNA. While these experiments are being planned for the future, we are not able, at this stage, to predict when we will have these results (the timeline prediction is even more uncertain due to the evolving COVID-19 related restrictions imposed in South Africa which has significantly impacted research). In the current study we therefore emphasize that the HIV-1-miR300c-3p-ZEB1/2 axis is a novel association which forms the basis for further future research. Furthermore, we have included a paragraph in the discussion section (lines 397 – 411) to emphasize the complexity of the factors involved in driving cellular migration, that our study shows an interesting association which will need to ne confirmed using loss- and gain-of-function studies.

Indeed, the rational for the focus on miR-200c-3p was not clear. We have elaborated upon this and the information can be found in the results section (lines 253-260).

  1. The association between hsa-miR-200c-3p and ZEB1 should be verified by the gain-or loss- function studies.

Author response: Several studies, using gain-or-loss of function experiments, have reported on the direct regulation of ZEB1 by hsa-miR-200c-3p, and these are displayed in Table 2. We agree that performing gain-or-loss of function studies within our own cells will allow us to further confirm this association and we have planned this in future studies. In the current study we report on a novel potential association of the downregulation of hsa-miR-200c-3p with enhanced migration in HIV exposed BL cells, and that this is potentially driven by enhanced expression of ZEB proteins.

  1. The authors indicated that hsa-miR-200c-3p serves as a novel biomarker for prognosis in patients with Burkitt lymphoma, who are HIV positive. However, there is no clinical data to support this conclusion.

Author response: We indeed report hsa-miR-200c-3p as a potential biomarker, based on our result, and indicate that this will have to be validated in clinical studies (lines 22-24 and 413-415).

Minor points

  1. Figure 2 information is not complete.

Author response: We apologise for this oversight. We have adjusted the Figure such that it now fits within the page.

  1. In line 75-76, “migration in invasion” is elusive

Author response: This has been corrected to read “migration and invasion” (Now Line 78).

  1. In line 246 "has-miR-200c-3p"

Author response: Thank you, we have corrected this (Now line 277).
